# communications
# earth & environment

# Ultra-low-velocity anomaly inside the Pacific Slab near the 410-km discontinuity

Jiaqi Li [1,2✉], Thomas P. Ferrand [3], Tong Zhou[1,2,4], Jeroen Ritsema [5], Lars Stixrude [2] & Min Chen[1,6]

The upper boundary of the mantle transition zone, known as the "410-km discontinuity", is attributed to the phase transformation of the mineral olivine (α) to wadsleyite (β olivine). Here we present observations of triplicated P-waves from dense seismic arrays that constrain the structure of the subducting Pacific slab near the 410-km discontinuity beneath the northern Sea of Japan. Our analysis of P-wave travel times and waveforms at periods as short as 2 s indicates the presence of an ultra-low-velocity layer within the cold slab, with a P-wave velocity that is at least ≈20% lower than in the ambient mantle and an apparent thickness of ≈20 km along the wave path. This ultra-low-velocity layer could contain unstable material (e.g., poirierite) with reduced grain size where diffusionless transformations are favored.

[1] Department of Computational Mathematics, Science and Engineering, Michigan State University, East Lansing, MI 48824, USA. [2] Department of Earth, Planetary, and Space Sciences, University of California, Los Angeles, CA 90095, USA. [3] Institüt für Geologische Wissenschaften, Freie Universität Berlin, Malteserstraße 74-100, Berlin 12249, Germany. [4] Aramco Research Center, Beijing–Aramco Asia, Beijing 100102, China. [5] Department of Earth and Environmental Sciences, University of Michigan, Ann Arbor, MI 48109, USA. [6] Department of Earth and Environmental Sciences, Michigan State University, East Lansing, MI 48824, USA. ✉email: jli@epss.ucla.edu

Under equilibrium conditions, the mineral olivine (α) transforms to wadsleyite (β olivine) near 410 km depth[1,2]. The phase transition is a global boundary known as the "410-km discontinuity", where the seismic wave speed increases by 3–5 percent[3–7]. Near cold subducting slabs (e.g., with a temperature lower than 1000 °C), the 410-km discontinuity can be elevated due to the positive Clapeyron slope of the α-β phase transition[8]. Inside the harzburgitic layer of the slab (e.g., with a temperature lower than 500–600 °C near its center), the nucleation and growth mechanism is inhibited[9,10]. New pathways in the form of shear transformation mechanisms have been supported by the recent discovery of a new shear-induced high-pressure olivine polymorph, the ω-olivine phase (also known as ε*-phase), in heavily-shocked meteorites[11–16]. If this new diffusionless process occurs in Earth's mantle, it will cause the seismic wave speed in the cold slab to decrease during mineral destabilizations associated with significant grain size reduction at low temperatures. Such a scenario could delay the arrival times of seismic waves that travel a long distance through the slab in the vicinity of the α-β phase transition.

In this paper, we analyze P waves generated by three earthquakes that occurred near the Kuril Islands and were recorded by seismic stations in northeastern China. The position of the Kuril earthquakes to the seismic stations is ideal for studying the structure of the Pacific slab near the 410-km discontinuity because P waves propagate hundreds of kilometers above or through the slab as illustrated by the Slab2.0 model[17] (Fig. 1a) and by the high-resolution tomography model FWEA18[18] of the mantle beneath the northwestern Pacific (Fig. 1b, c). In addition, the seismic stations are at epicentral distances between 1700 and 2400 km and record the triplication of P waves. The waveforms have up to three distinct pulses related to the direct wave that propagates above the 410-km discontinuity and the upper interface of the slab and refracted waves that cross them. The 700-km long array aperture and the 50-km average station spacing allow us to link the traveltime and waveform

characteristics of the P-wave triplication to underground structures at a ~18-km resolution scale (i.e., corresponding to the width of the Fresnel zone for a P wave with a dominant period of 2 s).

We particularly focus on an intermediate-depth earthquake (i.e., event 20091010, which occurred on October 10, 2009, Mw 5.9) since there are least interferences from its depth phases due to the relatively deeper focal depth of 114 km. We separate the seismic data into two subsets. The waveforms from stations along the source-receiver path RR' are the reference data. Along RR' the upper interface of the Pacific slab is located at about 450 km depth well below the 410-km discontinuity (Fig. 1b). Since P waves along RR' turn above the slab interface (the ray-tracing is based on a fast-matching method[19] with model FWEA18), we expect that the behavior of the P-wave triplication can be explained by standard 1-D seismic velocity profiles of the mantle (Supplementary Fig. 1). In contrast, P waves recorded by stations along SS' to the south of RR' propagate several hundreds of kilometers through the high-velocity slab and are complicated by layering in the slab in the vicinity of the 410-km discontinuity (Fig. 1c).

## Results and discussion

**Seismic observations.** The onset times of the refracting P waves (Fig. 2a) provide the first clue that P waves along SS' have been delayed by a low-velocity zone within the slab. First, the traveltime gradients indicate that the apparent velocities of the refracted waves are smaller for SS' (i.e., 9.89 km/s) than for RR' (i.e., 10.77 km/s) even though they have propagated within the high-velocity slab along SS'. Second, the travel times of the refracted waves along SS' are similar or up to ~1 s longer than refracted waves along RR' for epicentral distances larger than 1800 km (Fig. 2a). For RR', the refracted waves cross the 410-km discontinuity in the "normal" mantle (i.e., above the slab interface). However, for SS', the refracted waves cross the 410-km

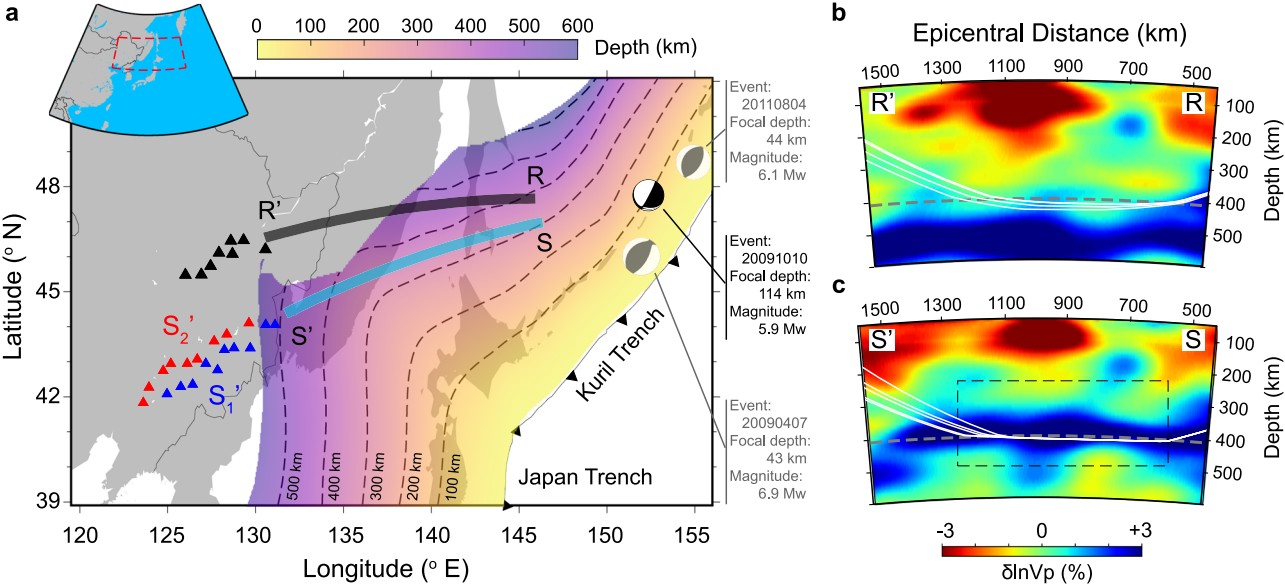

**Fig. 1 Research region and cross sections. a** The study area where the black beach ball indicates the epicenter of the intermediate-depth earthquake analyzed in this study, the triangles mark the locations of the seismic stations (black for RR', blue for SS₁', and red for SS₂') along paths RR' (black), and SS' (cyan). The gray beach balls denote the two shallow events also analyzed in this study, and the dashed contour lines with yellow-to-purple colors indicate the depth of the upper interface of the Pacific slab according to the Slab2.0 model[17]. **b** Cross section of tomography model FWEA18[18] for the reference region RR', the solid white curves are ray paths for the refracted waves that turn below 410 km and above 450 km and are recorded by stations at epicentral distances between 1700 and 1900 km (using a fast-matching ray-tracing method[19] with model FWEA18). The dashed gray line denotes the location of the 410-km discontinuity in the IASP91 model[6]. **c** Cross sections for the southern region SS'. The dashed box shows the 600-km wide and 250-km thick region where the wave speed structure is estimated by waveform modeling.

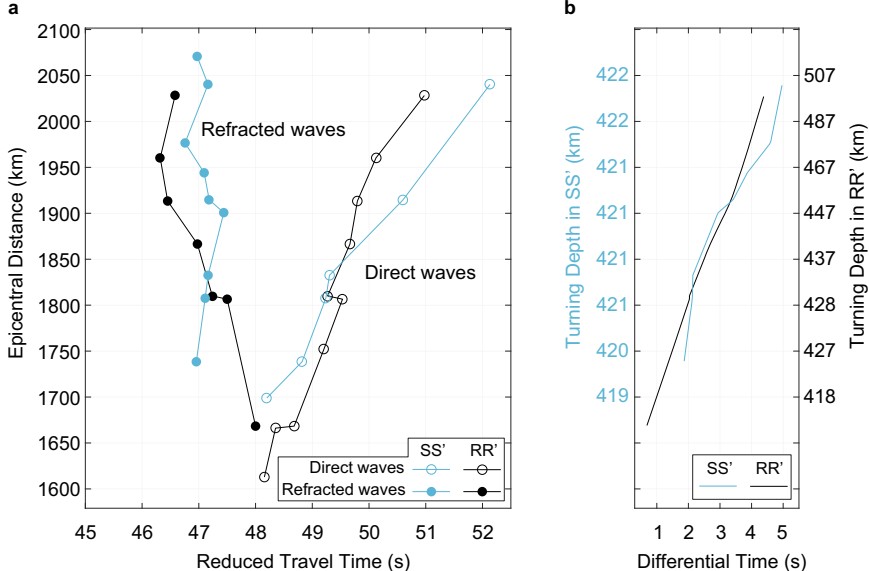

**Fig. 2 Travel time analysis. a** Arrival times of the first-arriving refracted wave (solid circles) and the direct wave (open circles). The measurements are shown in black, and cyan for the RR' and SS' paths, respectively. See also Supplementary Fig. 2. Note a reduced velocity of 10.1 km/s is applied to the time axis. **b** The traveltime difference between the direct wave and the refracted wave. The y-axis indicates the turning depth of the refracted wave, converted from the corresponding epicentral distance in (**a**) using a fast-matching ray-tracing method[19] with model FWEA18 for SS' to the left in cyan and RR' to the right in black.

discontinuity and propagate horizontally within the slab. Therefore, the similar travel times of the refracted waves point to a low-velocity zone inside the slab.

An alternative explanation for the delayed arrival time of the refracted waves along SS' can be the lower velocities near the stations (e.g., at an epicentral distance of about 1500 km in Fig. 1c). Since such velocity anomaly will affect the direct waves and refracted waves to the same extent due to their almost identical ray paths at shallower depths (see the delayed arrival times for the direct waves along SS' in Fig. 2a), the differential travel time between the refracting and direct P waves can cancel out such influence. The almost identical differential travel times between SS' and RR' (Fig. 2b) confirm our speculation of a low-velocity zone inside the slab.

By comparing recorded and computed waveforms of the P waves, we constrain the location, thickness, and velocity reduction of this low-velocity anomaly. We compute waveforms up to 0.5 Hz using the 2-D finite-difference method[20] to avoid the high computational cost of 3-D solvers at 0.5 Hz. However, we acknowledge that complex 3-D wave propagation along the slab interface may cause some of the complexity in the P waveforms. We use the FWEA18 tomographic model of the mantle as a reference structure and perturb the layered wave-speed structure near the P-wave turning points (the region indicated by the box in Fig. 1c) to better explain the waveform. We argue that our quasi-2-D modeling of P-wave perturbations is justified given the 1-D character of the wave speed anomalies in FWEA18 near the P-wave turning points, although more complex structures away from the turning points (e.g., near the source region) could contaminate the results (see Methods). We compare the synthetic and recorded waveforms after aligning each trace on the first arrival to isolate the influence of heterogeneity near the P-wave turning points on the travel time differences and amplitudes of the direct and refracted waves. Waveforms are band-pass filtered between 0.02 and 1 Hz to minimize side-lobes. Since the waveforms with ray paths sub-parallel to the strike direction are very sensitive to the depths of the subducting slab, we further divided the stations along SS' into

sub-linear arrays SS₁' and SS₂' for source-station azimuth ranges of 262−264° and 264−266°, respectively.

There are two significant differences between the recorded and the synthetic P wave triplication computed for FWEA18 for the SS₁' and SS₂' profiles of event 20091010 (Fig. 3a, c). For epicentral distances smaller than 1850 km, when the P waves propagate near the upper surface of the slab, the recorded P wave is weaker than the computed P wave because its energy is divided into two separate signals (see the waveforms for stations DNI, NE5E, NE4A along SS₁' and station NE5D along SS₂'). Between 2000 and 2200 km, when the P waves propagate through the slab, the recorded P waveforms have three distinct peaks near about 46 s, 49 s, and 52 s. The FWEA18 synthetics match the first and the third peak (see for example stations HST, JCT, and LHTJ along SS₁' and NE3A and PST along SS₂') but do not predict the presence of the second peak due to the P-wave refraction through the slab's upper surface. The waveform mismatch discrepancy at 0.5 Hz between the FWEA18 synthetics and the recorded waveforms is not surprising because the FWEA18 image is developed from waveforms with frequencies lower than 0.125 Hz (Supplementary Fig. 3) and has been parameterized to accommodate velocity variations over spatial scales larger than 50 km[18]. In contrast, along RR' where seismic waves propagate in the ambient mantle (i.e., above the slab), the recorded waveforms are simpler with at most two peaks (Supplementary Fig. 4c), and the mismatches between the recorded data and the FWEA18 synthetics are mainly due to arrival time differences between the two peaks.

**Ultra-low-velocity anomaly inside the slab.** Our preferred seismic model, called "FWEA18-LVZ", is a modification of FWEA18 in mainly two ways (e.g., see Fig. 4a). First, the strength of the wave-speed anomaly in FWEA18 is increased from +2% to +6%. This reduces the arrival time misfit, and the stronger wave-speed contrast across the slab's upper interface predicts the double-peaked P waveform for the nearest stations (<1850 km). The slab's P-wave velocity anomaly of +6% in FWEA18-LVZ (increased from +2% in FWEA18), to the ambient mantle, is consistent with previous waveform modeling studies of the

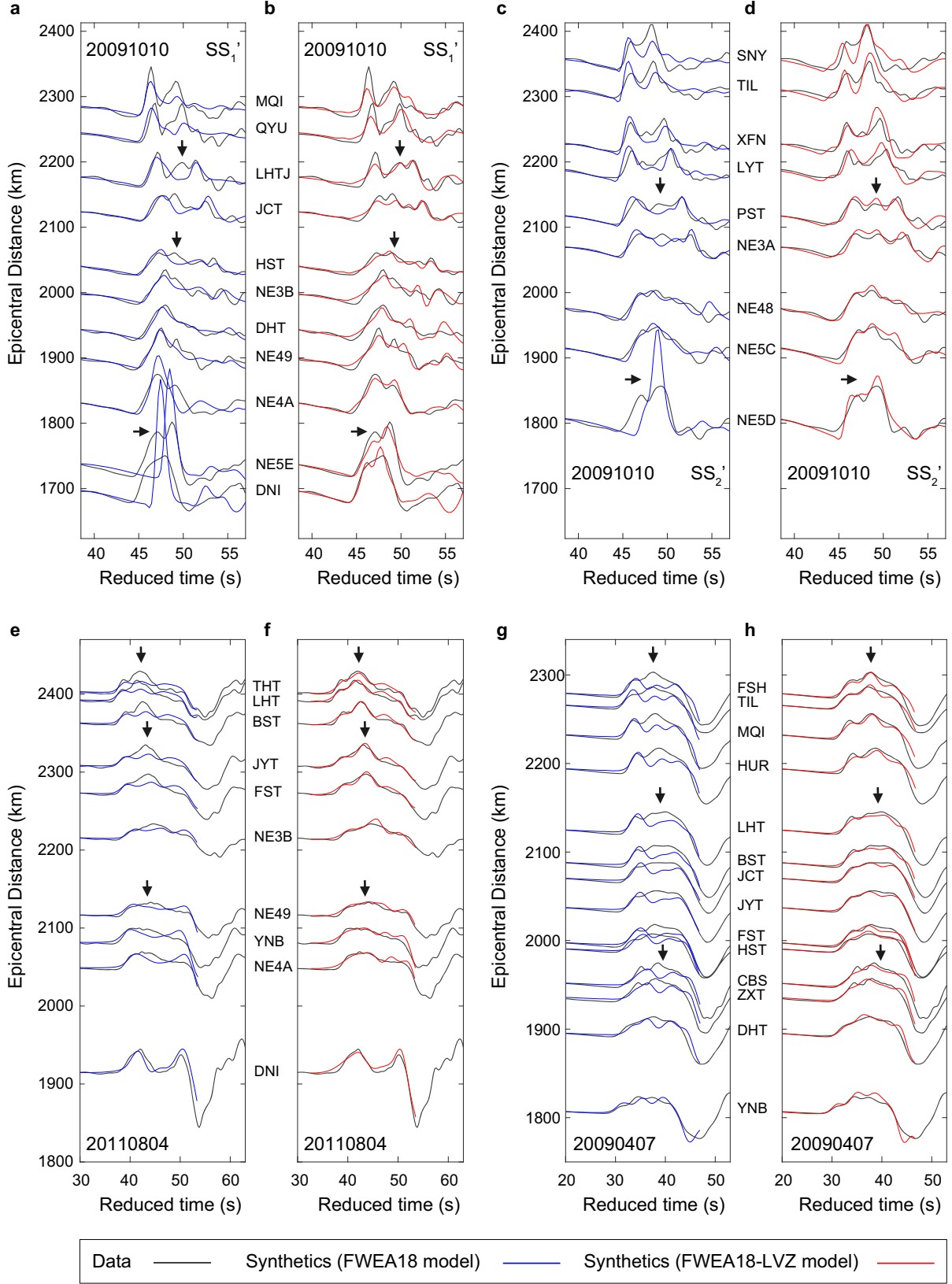

northeastern Pacific region[21,22]. Second, a low-velocity layer centered at around 400 km depth (390–395 km along $SS_1$' and 402–412 km along $SS_2$') predicts the second of the three P-wave signals recorded between 2000 and 2200 km that is missing in the FWEA18 synthetics. The P-wave speed reduction must be at least ≈20%, with an apparent thickness of up to ≈20 km along the wave path (Fig. 4a, b). The length of the low-velocity layer is fixed at 600 km, based on the length of the P wave segments through the slab along the $SS_1$'and $SS_2$' profiles (see Fig. 1). The low-velocity layer decelerates the direct wave traveling through the slab and it extends the branch (cusp) of the triplication, generating a second pulse in the waveform (see Supplementary Fig. 5). Only a

**Fig. 3 Comparison between the recorded and synthetic waveforms. a, b** Comparison between the recorded (black) and synthetic waveforms (blue and red) for paths SS₁' of event 20091010. The blue waveforms in (**a**) are calculated for the FWEA18[18] model and the red waveforms in (**b**) are computed for model "FWEA18-LVZ" derived in this study (see Fig. 4) using a finite-difference simulation code[20]. The horizontal arrows mark amplitude mismatches between the recorded and FWEA18 waveforms. The vertical arrows mark the presence of a second P-wave pulse. **c, d** Similar comparison for paths SS₂' of event 20091010. **e, f** Similar comparison for event 20110804. **g, h** Similar comparison for event 20090407. Note that a reduced velocity of 10.1, 9.8, and 9.7 km/s has been applied for events 20091010, 20110804, and 20090407, respectively. The synthetic waveforms of the depth phases (i.e., the large negative pulses in e-h) are not shown for the shallow events.

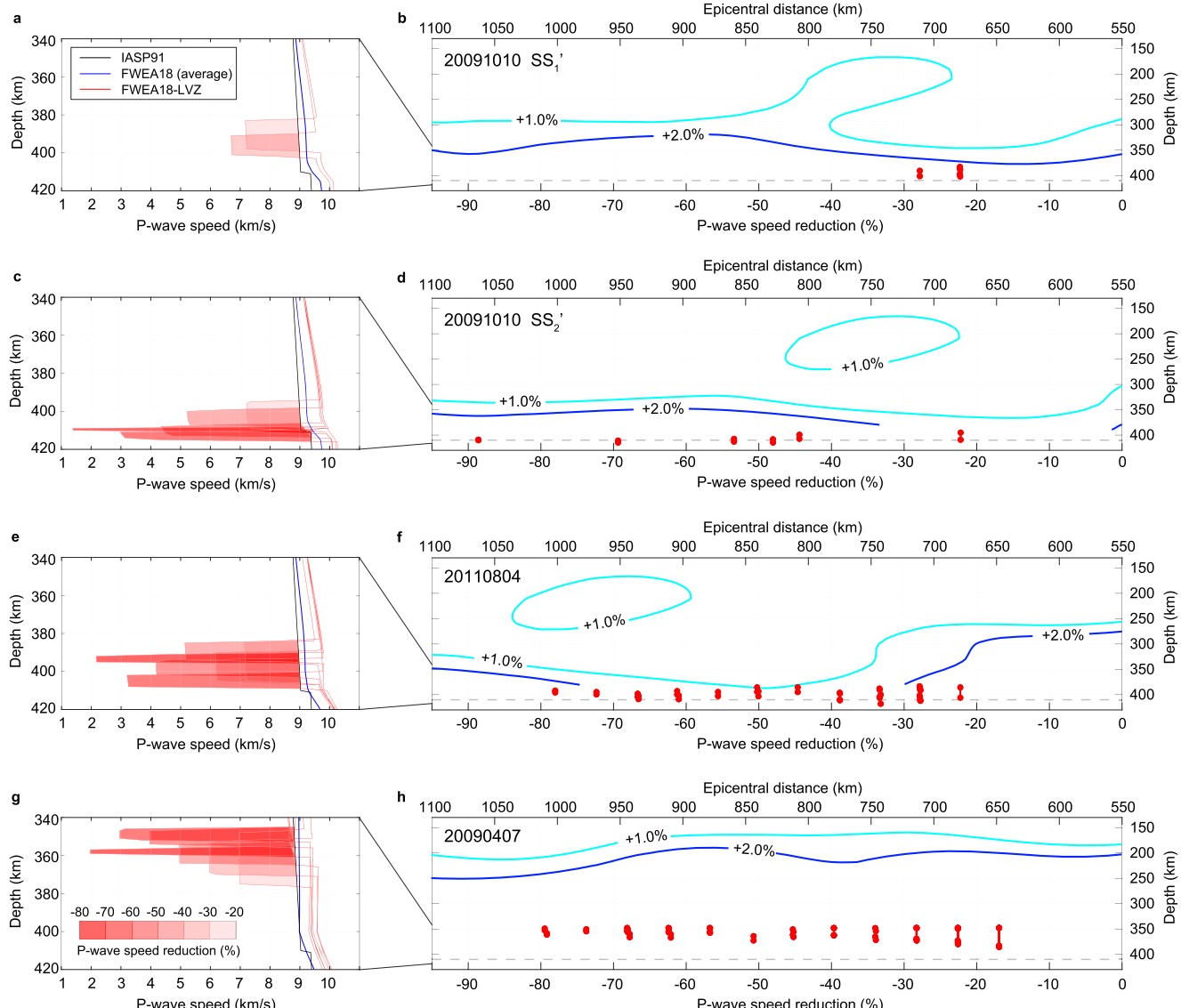

**Fig. 4 Acceptable models from waveform modeling. a** The black line marks the IASP91 model[6], and the blue line shows the average (in the dashed box of Fig. 1c) value of the FWEA18 model[18] for the SS₁' region. The red lines present typical acceptable models derived in this study for path SS₁' of event 20091010, and the color shading indicates the amplitude of the P-wave speed reduction (in %). **b** Comparison between the locations of the derived ultra-low-velocity zone and the upper surface of the subducting slab for path SS₁' of event 20091010. The red vertical bars with dots at both ends indicate the locations (and the layer thickness) and P-wave speed reductions (in %, bottom horizontal axis) of the ultra-low-velocity zone for the acceptable models (e.g., in **a**). The cyan and blue curves are the P-wave speed perturbation contours (i.e., +1.0% and +2.0%, respectively) from the FWEA18 model near the upper surface of the subducting slab, corresponding to the horizontal axis at the top. Note that the locations of the ultra-low-velocity zone and the slab contours share the same vertical axis (i.e., depth), but there is no correspondence between the horizontal axis at the top and the horizontal axis at the bottom. **c, d** Similar comparison for path SS₂' of event 20091010. **e, f** Similar comparison for event 20110804. **g, h** Similar comparison for event 20090407.

low-velocity layer (rather than a high-velocity discontinuity) can generate the missing second phase while maintaining the travel time difference of ~6 s between the first (at ~46 s) and the third (at ~52 s) pulses since the average wave speed in the slab is higher in FWEA18-LVZ than in FWEA18.

We explore the thickness and the velocity reduction of the low-velocity layer by a hybrid modeling approach. For each value of the velocity reduction from −0.5 km/s to −8.0 km/s (with intervals of 0.5 km/s), we determine the optimal thickness of the low-velocity zone and the vertical velocity structure (inside the box in Fig. 1c),

via the Niching Genetic Algorithm[23,24] (see Methods), to minimize the differences between recorded and synthetic waveforms. Figure 4c, d show all the acceptable models that fit the recorded waveforms equally well. For instance, the match between the recorded and computed waveforms is similar for a low-velocity layer with either a thickness of ≈20 km and a velocity reduction of ≈20% or a thickness of ≈2 km and a velocity reduction of ≈90%.

The nonuniqueness of the velocity reduction and the layer thickness is due to the minimum spatial resolution since the Fresnel zone of a P wave (~0.5 Hz) is ~18 km wide near the 410-km discontinuity for event 20091010 with a magnitude of 5.9 (Mw). This trade-off is evident in the acceptable models in the last iteration (Fig. 4), and also in the posterior distribution of all the 1,280 models (see Supplementary Fig. 6).

By experimentation, we find that the estimates of the thickness of the low-velocity layer and its velocity reduction slightly depend on the width of the low-velocity layer. However, a minimum width of 200 km is necessary to produce the second P-wave signal (Supplementary Fig. 7). The existence of such a low-velocity layer inside the slab is not only identified by the non-gradient-based modeling approach (with the Niching Genetic Algorithm[23,24]) but also confirmed by the sensitivity kernels derived from the adjoint method[25] (Supplementary Fig. 8). Finally, our results would not have changed if we had chosen a different reference seismic structure (i.e., the 1-D IASP91 model instead of the FWEA18 model): although the waveform fits are worse, a low-velocity layer centered at 370 km depth with a velocity reduction of 10–20% and a thickness of 20 km also explained the essential waveform features (Supplementary Fig. 9).

The observed P-wave phenomenon should be a robust feature reflecting the ultra-low-velocity layer inside the slab, rather than artifacts from either the earthquake or the local structures beneath the stations. To further validate, we examine two more shallow events (i.e., 20090407 to the south and 20110804 to the northeast) in the same subduction zone but with different epicentral locations and focal depths (Fig. 1a). Both new events exhibit the absence of a second peak in the FWEA18 synthetics (indicated by the vertical arrows in Fig. 3e, g) as shown in the studied intermediate-depth event 20091010 (Fig. 3a, c). Similarly, the "FWEA18-LAZ model" (i.e., with a P-wave velocity reduction of at least ≈20% inside the slab, see Fig. 4e, g) can reproduce the occurrence of a second peak (Fig. 3f, h).

**Origin of the anomaly.** Partial melting is inferred from the transition-zone water filter hypothesis[26] and it is reasonable to consider it as a possible explanation for the wave speed anomaly observed above the 410-km discontinuity, as usually done to interpret low-velocity layers in equilibrated mantle regions[27,28]. However, there is no evidence for such a low velocity between 383 and 415 km depth for the reference source-receiver path RR' where the slab is below the 410-km discontinuity (Supplementary Fig. 1). Therefore, it appears that the resolved low-velocity zone is confined to the subducting slab, rather than caused by the presence of melt above the 410-km discontinuity (otherwise, we should expect to observe it along RR' as well). Thus, partial melting unlikely explains the velocity anomaly, as further confirmed by the fact that the location of the ultra-low-velocity layer along SS$_1$' (383–402 km) is shallower than that along SS$_2$' (395–415 km), well correlating with the relative depths of the slab depth contours in these two regions (Fig. 1a).

Further comparisons with the P-wave velocity perturbations from the tomographic model FWEA18 show that the locations of the derived ultra-low-velocity layer (in cross sections for events 20091010 and 20110804, see Fig. 4a–f) are close to the +2.0% perturbation contour (i.e., near the slab upper surface) and in the

vicinity of the 410-km discontinuity. Therefore, this ultra-low-velocity layer could either reflect the basaltic layer of the slab or relate to the olivine phase transformations near the 410-km discontinuity. We prefer the latter interpretation because, in the cross section to the south where the subducting slab is at a shallower depth (i.e., for event 20090407, see Fig. 4g, h), the locations of the ultra-low-velocity layer are no longer close to the +2.0% perturbation contour. Though the location of the ultra-low-velocity-layer in this cross section is relatively less constrained because the top half of the slab is at shallower depths (i.e., ~200–350 km) with sparse seismic paths coverages, the long distance (~100 km) between the +2.0% contour and the ultra-low-velocity layer indicate that such a layer is likely inside the depleted harzburgitic layer of the slab, rather than in the basaltic crust.

The velocity reduction in this layer is much stronger than the P-wave velocity contrast of 2–6% between the slab (near its cold core, i.e., close to the center of the blue block in the tomography image) and the ambient mantle[18,21,22]. Our waveform modeling indicates that a −6% velocity reduction is not sufficiently strong to produce a second P wave signal. Therefore, we rule out the possibility that a slab gap[29] could explain the low-velocity layer. Moreover, there is neither evidence for a gap in the Slab2.0 model nor tomographic evidence that the Pacific slab would be weak and detached near the P-wave turning points, i.e., near 139° E longitude and 46° N latitude (see Fig. 1a). The presence of metastable olivine in cold regions, can also lower the P-wave velocity about −2% to −5% than the normal mantle, as reported by several seismological studies to the southwest of our study region[30–32]. However, the velocity reduction up to −5% is not sufficient to reproduce the seismic waveforms. Therefore, metastable olivine alone is not a viable explanation for the significant velocity reduction of at least ≈20%.

Phase transformation, with volume reductions, is a mechanism for the reduction of the effective bulk modulus[33,34]. In zones (e.g, thickness of 5–30 km[35–37] for the 410-km discontinuity) where olivine (low-pressure phase) and wadsleyite (high-pressure phase) coexist, the pressure perturbation caused by the passing of the seismic wave (e.g., on the order of $\sim 10^{-7}$ GPa)[38] will disrupt the equilibrium and induce the phase transformation. For a pyrolytic Earth model, the theoretical P-wave velocity reduction caused by the softening of the effective bulk modulus can be at most ≈30%[37,38], which falls well into our estimated value between 17% and 90% (Fig. 4). However, such softening of the elastic modulus is unexpected in the cold slab, due to the slower kinetics[39]. We observe the ultra-low-velocity zone only inside the slab (along SS') but not in the ambient mantle (along RR') where softening is more likely.

Our preferred explanation of this ultra-low-velocity zone inside the slab is the existence of a layer of destabilized olivine which is associated with a significant grain size reduction: spineloid olivine (β- and/or ω-olivine and/or other sheared spineloid phase) would form as lamellae discordant with the host olivine crystal, disorganizing the structure homogeneity, and yielding a seismic wave speed reduction. Subsequent instabilities of grain boundaries are expected to maintain small grain sizes within the transformation loop[40,41]. In an experiment with ultrasonic interferometry and in-situ X-ray diffraction, a reduction of shear-wave velocity (i.e., −7.5% to −1.8%) has been observed at the onset of the olivine-wadsleyite transformation and is interpreted as resulting from the existence of an intermediate spineloid phase (ω-olivine or similar spineloid phase)[40].

In addition to a P-wave (compressional wave) velocity drop, we have also observed a reduction in the S-wave (shear-wave) velocity (Supplementary Fig. 10). However, partly due to the larger attenuation of the S-waves at high frequency, the signal-to-noise ratio of the S-waves is relatively low, and the waveform fitting is not

as good as the P waves (Supplementary Fig. 11), resulting in a less constrained S-wave velocity drop of −85% to −11%.

Discrepancies between the experimental measurement (i.e., at most −7.5% for the S-wave velocity) and the one in this seismological study (i.e., at least −17% for the P-wave velocity, and at least −11% for the S-wave velocity) could be due to the distinct frequency ranges (i.e., MHz for the ultrasonic wave and Hz for the teleseismic wave) and/or potentially complex 3-D wave propagation. In addition, the maximum reduction of −7.5% in the experiment could be underestimated if the transformation is localized in the sample instead of homogeneously occurring. Nevertheless, we propose that the ultra-low-velocity layer we have identified could consist of an intermediate phase transiently existing during the olivine–wadsleyite phase transformation within the cold subducting slab, where diffusion-assisted processes are inhibited. This intermediate phase, known either as ε*-phase or ω-olivine[11–16], was at first theoretically predicted[11], then observed in meteorites[15,16]. It was recently named poirierite[16] and its transient (meta)stability could impact the rheology of the mantle[40,41]. Our seismological observations could highlight the transient (meta)stability of poirierite, under substantial shear stress, within the cold sinking slab. Given the discovery of the metastable olivine in the nearby subducting slab near Honshu[32], and the similar thermal parameters of the Honshu and Kuril arcs[42], the core of the subducting Kuril slab is sufficiently cold to inhibit the nucleation and growth mechanism. Nonetheless, any phase transformation at relatively low temperatures necessarily induces long-lived grain size reduction[43,44], which is also known to reduce P-wave velocity[45] and is expected to contribute to the observed reduction. The existence of an unstable (transforming) rim around the (meta)stable olivine wedge would also be supported by the high seismicity anomaly recently observed[46].

The ultra-low-velocity zone (centered around 400 km, with a wave speed reduction of at least ≈20%) is identified inside the subducting slab. The presence of the spineloid transient phase would facilitate the olivine transformations and thus maintain a localized transformation loop[40], which might solve the paradox between a gradual 410-km discontinuity (i.e., 7–19 km) predicted by thermodynamic and mineralogical models[47,48] and a sharp interface discovered by seismologists (e.g., 2–10 km near and inside the Tonga slab[49]). The poirierite structure by the shear mechanism, as a new pathway, would also enhance the olivine-wadsleyite transformation at high-stress (e.g., in shocked meteorites) and low-temperature conditions (e.g., in subducting slabs)[15,16]. Further theoretical and experimental studies are needed to examine the value range we have provided in this study and to evaluate how the grain size is controlled by grain-boundary instability, poirierite or any other spineloid transient phase within the transformation loop. Global-scale seismic studies based on both the P and S waves and 3-D modeling are needed to provide more insights into olivine phase transitions in the deep Earth and planetary materials.

## Methods

**Data processing**. The waveform data used in this paper are recorded by the broadband stations from the permanent CEArray[50] and the temporary NECESSArray (the NorthEast China Extended Seismic Array) in northeast China. The location, magnitude, and focal mechanism of the earthquakes are from the Global Centroid Moment Tensor Project[51].

We applied a Butterworth filter (first-order, zero-phase shift) with a frequency band from 0.05 to 1 Hz to the displacement data, after removing the instrument response. For a better illustration of the waveform (Fig. 3), we applied a reduced velocity to the time axis to allow stations at different epicentral distances to appear in the same time window.

**Waveform inversion**. Since triplications have the advantage of minimizing the influence from the shallower part and emphasizing structures near the turning points of the ray paths, researchers[52,53] used a 1-D model throughout the research region. In this study, to increase the accuracy, we chose the 2-D tomographic model FWEA18[18] as the background P-wave model in a certain cross-section.

During the inversion, we kept the model outside of the inversion box (dashed box in Fig. 1c) unchanged, and only modified the region inside the box by adding 1-D model perturbations. Since we only modify the velocity structure of FWEA18 near the P-wave turning points, unconstrained parts of FWEA18 (with probably underestimated wave speed) near the source region might lead to some overestimation of the inverted values near the turning points. Nevertheless, the current P-wave velocity anomaly in the FWEA18 model is not large enough to predict the double-peaked P wave.

For the model setup, for anchor points away from the interface, we set the interval to be 20 km and allowed the P-wave speed at each anchor point to change within ±0.5 km/s. In addition, there are also anchor points immediately on the discontinuity to capture the velocity jump. The location of the interface and thickness are also free parameters, with a relatively large variable range of ±35 km and 0–50 km, respectively. The variable range of the velocity in the ultra-low-velocity zone will be discussed in the following 'tradeoff' section.

The inversion code is based on the non-gradient-based Niching Genetic Algorithm[23,24], which searches the model space via massive forward modeling. As for the misfit window, we choose a continuous window from 39 to 57 s (reduced time). We note that before calculating the L2 norm of the differences between the data and synthetics, we aligned them by cross-correlation to emphasize the relative information between triplicated waveforms and to mitigate baseline shifts.

We selected a GPU-based 2-D finite-difference method[20] for our forward modeling tool to reduce the computational costs for the non-gradient-based inversion approach. For one 2-D simulation (up to 1 Hz), the computation takes 6 s on one NVIDIA V100 GPU card. We note that several corrections, e.g., out-of-plane, point source excitation, and Earth-flattening, have been incorporated to better account for the 3-D wavefield spreading[20].

**The tradeoff between model parameters**. To avoid falling into local minima, in the inversion code[24], we divided the model population into subpopulations. And a penalty term for the model similarities between subpopulations is applied. In each inversion, we performed 100 iterations. And in each iteration, there are 80 models in 10 subpopulations, to ensure model diversity. In total, we searched 8000 models in each inversion.

To explore all possible candidates in the model space, for each region, we performed 16 inversions, instead of one. For each inversion, the location and thickness of the low-velocity layer are still free parameters. The only difference is that we forced the velocity reduction to be fixed (i.e., from −0.5 km/s to −8.0 km/s with intervals of 0.5 km/s) to reduce the total number of free parameters at the expense of performing more inversions. With randomly distributed model parameters as inputs, most of the models in the last iteration are clustered and follow a curve (see Supplementary Fig. 6), indicating the trade-off between velocity reduction and layer thickness. Nevertheless, the product of these two parameters is constant (Supplementary Fig. 6). Finally, we further examined the waveform fitting of all the models in the last iteration and selected the final acceptable models (Fig. 4).

**Robustness of the ultra-low-velocity layer**. For the inversion region (dashed box in Fig. 1c), the vertical height is about 250 km, which is large enough (about twice the thickness of the slab). In the horizontal direction, we set the length to be 600 km, which contains most of the ray paths near the turning points. To test how the inversion region affects the results, we conducted extra inversions with different horizontal lengths of 200 km, 300 km, 400 km, and 800 km, respectively. Results show that when the length is too short (e.g., 200 km), no model can fit the data. For other lengths, the inverted models are consistently pointing to the existence of the ultra-low-velocity zone, with slightly different locations and amplitudes (Supplementary Fig. 7).

To test the dependency of our results on the initial model of FWEA18, we performed another 1-D inversion. In this case, we chose the QSEIS[54] code as the forward modeling tool. In the inverted 1-D model sets, a significant velocity reduction (−10% to −20% when the layer thickness is about 20 km) is also observed (Supplementary Fig. 9).

Besides the non-gradient-based inversion approach, we also used SPCFEM2D[25] to calculate the sensitivity kernels. In practice, we calculate the adjoint source for certain stations (NE3A, PST, and LYT for the northern region. HST, JCT, LHTJ, and QYU for the southern region). With the adjoint sources, we run the adjoint simulation and then get the misfit function gradient, also known as the sensitivity kernel (Supplementary Fig. 8).

## Data availability

The seismic data generated during the analysis, i.e., both P- and S-wave displacement records after removing the instrument responses, are available as Supplementary Data 1 and also uploaded on the Zenodo repository[55]: https://doi.org/10.5281/zenodo.7655059.

## Code availability

All the computations made in this paper are either described in the method section or based on codes that are cited in the reference list. We also put together the inversion

codes used in this paper as Supplementary Software 1 and also on the Zenodo repository[56]: https://doi.org/10.5281/zenodo.7655063.

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

## Acknowledgements

We thank M.C. (passed away on 2021 July 18), who built the connection between J.L. and T.P.F. through an online seminar, during the COVID-19 pandemic. Seismic records used in this study came from the CEArray and the NECESSArray, and we thank the team members for their deployments. We thank T. Bao, J. Ning, S. van der Lee, Z. Xi, and X.

He for discussions. We thank the Institute for Cyber-Enabled Research (ICER) at Michigan State University and the Extreme Science and Engineering Discovery Environment (XSEDE supported by National Science Foundation [NSF] Grant ACI-1053575) for providing high-performance computing resources. T.P.F. acknowledges the *Alexander von Humboldt* Foundation (Germany) and T.Z. acknowledges the National Scientific Foundation of China (Grant No. 42276049). This research was supported by the NSF grant 1802247 and startup fund of M.C. at Michigan State University, and supported by the National Science Foundation under grant EAR-1853388 to L.S.

## Author contributions

J.L. processed the seismic data and developed the inversion code. T.P.F., J.L., M.C., T.Z., and J.R. analyzed the results. T.P.F. proposed the preferred interpretation and discussed with J.L. and L.S. on the relevance of alternative explanations. J.L. and J.R. took the lead in writing the manuscript, and all authors discussed the results and edited the manuscript.

## Competing interests

All authors declare no competing interests.
