## [Peer Review File · Communications Earth & Environment]

This manuscript has been previously reviewed at another Nature Portfolio journal. This document only contains reviewer comments and rebuttal letters for versions considered at Communications Earth & Environment.

17th Feb 23

Dear Dr Li,

Please allow me to apologise for the delay in sending a decision on your manuscript titled "Ultra-low-velocity anomaly inside the Pacific Slab near the 410-km discontinuity". It has now been seen by our reviewers, whose comments appear below. In light of their advice I am delighted to say that we are happy, in principle, to publish a suitably revised version in Communications Earth & Environment under the open access CC BY license (Creative Commons Attribution v4.0 International License).

We therefore invite you to revise your paper one last time to address the remaining concerns of our reviewers. At the same time we ask that you edit your manuscript to comply with our format requirements and to maximise the accessibility and therefore the impact of your work.

EDITORIAL REQUESTS:

*****Please take care to match our formatting and policy requirements. We will check revised manuscript and return manuscripts that do not comply. Such requests will lead to delays. *****

SUBMISSION INFORMATION:

OPEN ACCESS:

Communications Earth & Environment is a fully open access journal. Articles are made freely accessible on publication under a [CC BY license](http://creativecommons.org/licenses/by/4.0) (Creative Commons Attribution 4.0 International License). This license allows maximum dissemination and re-use of open access materials and is preferred by many research funding bodies.

For further information about article processing charges, open access funding, and advice and support from Nature Research, please visit a

href="https://www.nature.com/commsenv/article-processing-charges">https://www.nature.com/commsenv/article-processing-charges

At acceptance, you will be provided with instructions for completing this CC BY license on behalf of all authors. This grants us the necessary permissions to publish your paper. Additionally, you will be asked to declare that all required third party permissions have been obtained, and to provide billing information in order to pay the article-processing charge (APC).

[link redacted]

Best regards,

Joe Aslin

Senior Editor,
Communications Earth & Environment
<https://www.nature.com/commsenv/>
Twitter: @CommsEarth

REVIEWERS' COMMENTS:

Reviewer #1 (Remarks to the Author):

The authors analyzed waveform triplication data to study velocity structure around the 410-km discontinuity within the subducting slab beneath northwest Pacific. They found the P waveforms from an intermediate depth earthquake and two shallow earthquakes recorded by broadband stations in northeast China have multiple arrivals, which cannot be explained by a regular 1-D velocity model.

They used a 2-D finite difference method to model the waveforms and a 1-D waveform inversion method to obtain velocity structures at the turning depths of the P waves. They found that the observed waveform complication is caused by an ultra-low-velocity layer within the subducting Pacific slab. The ultra-low-velocity layer is ~ 20 km thick and has a P-wave velocity reduction of ~17%. The layer extends ~300-800 km around the P-wave propagation direction with a width greater than 200 km. They speculate that the ultra-low velocity layer could be related to metastable ω -olivine (e.g., poirierite) with reduced grain size, which was discovered recently by mineral physics studies. I think the seismic observations here are very interesting and could have significant implications to our understanding of the mineral physics and dynamics of subducted slabs. Overall, the

manuscript is well written, and the seismic results are robust, therefore it is suitable to be published in Communication Earth & Environment with a minor revision. My detailed comments and questions for the authors are listed below.

1. Line 21: "Where temperatures are lower than 1000°C inside the harzburgitic..." need references
2. Line 40: "~15-km resolution scale..." I think the Fresnel zone is associated with a specific frequency.
3. Lines 45-46, Figure 1 caption: "... blue for S1S1' and red for S2S2' ..." I couldn't find the letters S1 and S2 in Figure 1(a).
4. Lines 49, Figure 1 caption: "...solid white curves are ray paths..." , 1-D ray paths based on the iasp91 model?
5. Line 106: "...band-pass filtered between 0.02 and 1 Hz..." The dominant period of the waveforms shown in Figure 3 is ~5 s. I am wondering why there are no high frequency (~0.2-1.0 Hz).
6. Line 192: "... the Fresnel zone of a P-wave (~0.5 Hz) is 15-20 km..." since the dominant period is ~5 s (0.2 Hz), I am wondering whether it is appropriate to use 0.5 Hz to compute the Fresnel zone.

Reviewer #2 (Remarks to the Author):

I am pleased to read the revised manuscript of Li and coauthors. The authors have made a good effort to answer my comments and suggestions, and those of the other reviewers. I think their revisions have really strengthened the paper. I therefore recommend publication of the paper.

Response to the editor and reviewers

[Manuscript COMMSENV-22-1326-T]

Dear Editor Dr. Aslin,

Thank you for your time and expertise with our manuscript. We appreciate the positive feedback from the reviewers. We have further improved our manuscript according to the constructive comments from Reviewer #1. We have also carefully followed the Editorial Requests Table to edit our manuscript to comply with the policy and formatting requirements of Communications Earth & Environment.

In the followings, the comments from the reviewers are written in **black** and **our responses in red**.

Reviewer #1 (Remarks to the Author):

The authors analyzed waveform triplication data to study velocity structure around the 410-km discontinuity within the subducting slab beneath northwest Pacific. They found the P waveforms from an intermediate depth earthquake and two shallow earthquakes recorded by broadband stations in northeast China have multiple arrivals, which cannot be explained by a regular 1-D velocity model.

They used a 2-D finite difference method to model the waveforms and a 1-D waveform inversion method to obtain velocity structures at the turning depths of the P waves. They found that the observed waveform complication is caused by an ultra-low-velocity layer within the subducting Pacific slab. The ultra-low-velocity layer is ~ 20 km thick and has a P-wave velocity reduction of ~17%. The layer extends ~300-800 km around the P-wave propagation direction with a width greater than 200 km. They speculate that the ultra-low velocity layer could be related to metastable-olivine (e.g., poirierite) with reduced grain size, which was discovered recently by mineral physics studies. I think the seismic observations here are very interesting and could have significant implications to our understanding of the mineral physics and dynamics of subducted slabs. Overall, the manuscript is well written, and the seismic results are robust, therefore it is suitable to be published in Communication Earth & Environment with a minor revision. My detailed comments and questions for the authors are listed below.

1. Line 21: “Where temperatures are lower than 1000°C inside the harzburgitic...” need references

A1: Thanks for this important comment. We admit that previously we did not clearly distinguish between the two different concepts of the temperature near the plate and its cold center. This time, we clarified this:

“Near cold subducting slabs (e.g., with a temperature lower than 1000°C), the 410-km can be elevated due to the positive Clapeyron slope of the α - β phase transition⁸. Inside the harzburgitic layer of the slab (e.g., with a temperature lower than 500-600°C near its center), the nucleation and growth mechanisms inhibited^{9,10}.”

Reference #8 is for the temperature of 1000°C and references #9-10 are for the temperature of 500-600°C.

We also added a sentence justify that the temperature of the subducting Kuril slab is sufficiently cold to inhibit the nucleation and growth mechanism:

“Given the discovery of the metastable olivine in the nearby subducting slab near Honshu (Shen and Zhan, 2020), and the similar thermal parameters of the Honshu and Kuril arcs (Syracuse et al., 2010), the subducting Kuril slab is sufficiently cold to inhibit the nucleation and growth mechanism.”

2. Line 40: “~15-km resolution scale...” I think the Fresnel zone is associated with a specific frequency.

A2: Thanks for pointing it out. We agree with the reviewer and clarified the corresponding frequency in the manuscript:

“(i.e., corresponding to the width of the Fresnel zone for a P-wave with a dominant period of 2 s).”

3. Lines 45-46, Figure 1 caption: “... blue for S1S1’ and red for S2S2’...” I couldn’t find the letters S1 and S2 in Figure 1(a).

A3: Thanks for this careful observation. To be more clear, this time, we replaced S1S1’ with SS1’, S2S2’ with SS2’ in Figure 1 caption and elsewhere to indicate that S1S1’, S2S2’, and SS’ share the same starting point, i.e., “S”.

4. Lines 49, Figure 1 caption: “...solid white curves are ray paths...”, 1-D ray paths based on the iasp91 model?

A4: The ray paths are based on FWEA18 with a 2-D ray tracing method (Ref 19). We clarified this in the figure caption.

5. Line 106: “...band-pass filtered between 0.02 and 1 Hz...” The dominant period of the waveforms shown in Figure 3 is ~5 s. I am wondering why there are no high frequency (~0.2-1.0 Hz).

A5: Thanks for this observation. I would like to clarify that the dominant periods of the waveforms in Figure 3e-h are ~ 5 s, however, the the dominant periods in Figure 3a-d are ~ 2 s:

1. For example, in Figure 3a, it is quite clear that the length of the waveform is ~ 2 s.
2. There are high frequencies in Figure 3a-d, but not in Figure 3e-h. This is because The magnitude of the event in Figure 3a-d is 5.9 Mw, however in Figure 3e-h is 6.1 Mw and 6.9 Mw. A larger magnitude will result in a longer source time function.

6. Line 192: “... the Fresnel zone of a P-wave (~0.5 Hz) is 15-20 km...” since the dominant period is ~5 s (0.2 Hz), I am wondering whether it is appropriate to use 0.5 Hz to compute the Fresnel zone.

A6: The dominant periods are ~ 2 s, 3 s, and 5 s for events 20091010, 20110804, and 20090407, respectively. Therefore, we agree with the reviewer that the appropriate size of the Fresnel zones for all the events should be from 18 km (event 20091010) to 45 km (event 20090407). In the manuscript, we clarified that the Fresnel zone size of ~ 18 km is for event 20091010 with the minimum magnitude:

“The nonuniqueness of the velocity reduction and the layer thickness is due to the minimum spatial resolution since the Fresnel zone of a P-wave (~ 0.5 Hz) is ~ **18 km** wide near the 410-km **for event 20091010 with a magnitude of 5.9 Mw.**”

Reviewer #2 (Remarks to the Author):

I am pleased to read the revised manuscript of Li and coauthors. The authors have made a good effort to answer my comments and suggestions, and those of the other reviewers. I think their revisions have really strengthened the paper. I therefore recommend publication of the paper.

A7: We thank the constructive and insightful comments and suggestions from all the reviewers. We also appreciate the revision process which significantly improved the quality of the paper.